# A High-Frequency Mechanical Scanning Ultrasound Imaging System

**DOI:** 10.3390/bios13010032

**Published:** 2022-12-27

**Authors:** Jie Xu, Ninghao Wang, Tianxiang Chu, Bingqian Yang, Xiaohua Jian, Yaoyao Cui

**Affiliations:** 1Academy for Engineering and Technology, Fudan University, Shanghai 200433, China; 2Suzhou Institute of Biomedical Engineering and Technology, Chinese Academy of Sciences, Suzhou 215163, China; 3School of Biomedical Engineering (Suzhou), Division of Life Sciences and Medicine, University of Science and Technology of China, Suzhou 215163, China; 4School of Energy and Power Engineering, Nanjing University of Science and Technology, Nanjing 210094, China

**Keywords:** high-frequency ultrasound, mechanical scanning, high imaging resolution

## Abstract

High-frequency ultrasound has developed rapidly in clinical fields such as cardiovascular, ophthalmology, and skin with its high imaging resolution. However, the development of multi-elements high-frequency ultrasonic transducers and multi-channel high-frequency ultrasound imaging systems is extremely challenging. Here, a high-frequency ultrasound imaging system based on mechanical scanning was proposed in this paper. It adopts the method of reciprocating feed mechanism, which can achieve reciprocating scanning in the 14 mm range at 168 mm/s with a small 60 MHz transducer. A single-channel high-frequency ultrasonic imaging system consisting of the transmitting module, analog front end, acquisition module, and FPGA control module was developed. To overcome the non-uniformity of mechanical scanning, the ultrasound images are compensated according to the motion trajectory. The wire target and ex vivo tissue experiments have shown that the system can obtain an imaging resolution of 51 μm, imaging depth of 8 mm, and imaging speed of 12 fps. This high-frequency mechanical scanning ultrasound imaging system has the characteristics of simple structure, high-frequency, real-time, and good imaging performance, which can meet the clinical needs of high-resolution ultrasound images.

## 1. Introduction

Ultrasound imaging is widely used in clinical practice because of its real-time, convenient, non-destructive, and radiation-free advantages. In particular, high-frequency ultrasound imaging, which can provide higher resolution images, has been rapidly applied and developed in recent years, such as ophthalmic ultrasound, skin ultrasound, intravascular ultrasound, ultrasound-guided catheterization, and so on [1,2,3,4,5].

Conventional ultrasound imaging systems mostly use multi-channel circuits and adapted multi-array transducers [6]. However, for high-frequency ultrasound imaging, this approach will make the system quite complex, bulky, and expensive. This is mainly due to the need for higher frequency electrical pulse transmitting modules, wider and flat system bandwidths, and high-speed ADC for data acquisition and transmission as the operating ultrasonic frequency increases. However, the maximum transmission frequency of the high-frequency transmitting chip currently on sale is only 35 MHz, and the maximum multi-channel ultrasonic acquisition frequency is 125 MHz, which cannot meet the requirements of high-frequency ultrasound system design exceeding 40 MHz. In addition, the fabrication of high-frequency multi-array ultrasonic transducers is also a huge challenge.

Single-element mechanical scanning imaging is an effective way for high frequency ultrasound imaging [7,8]. For example, clinical intravascular ultrasound imaging and endoscopic ultrasound imaging based on catheter 360-degree rotation imaging [9,10]. Some systems use mechanical scanning for high-frequency ultrasound imaging of skin and ophthalmic, but their maximum operating frequency is only about 20 MHz. Nevertheless, the use of higher frequency ultrasound imaging has become increasingly desirable in various applications, such as oncology, ophthalmology, and neurology.

Therefore, in this paper we designed a reciprocating mechanical scanning structure, made a miniature high-frequency ultrasonic probe, developed a single-channel high-frequency transmitting module and acquisition amplification module, and realized a 60 MHz high-frequency ultrasonic imaging system. To overcome the uneven characteristics of reciprocating mechanical scanning, a motion compensation method is proposed to correct the image position of the imaging data to achieve real-time precise scanning images. Finally, the system was used to perform ultrasound imaging experiments on the tungsten wire phantom and the fish eyeball, respectively.

## 2. Materials and Methods

### 2.1. System

The basic principle of our system is to use a high-precision mechanical scanning system to drive a single-element high-frequency ultrasonic transducer for real-time high-resolution ultrasound imaging.

#### 2.1.1. Mechanical Scanning System

This mechanical scanning system mainly adopts the method of reciprocating feed mechanism, which can avoid the dead point problem of the traditional connecting rod transmission scheme. The system is mainly composed of a stepper motor (20HS2806A4, SUMTOR), transducer base, crank, round slider, slotted link, linear bearing, and bracket, as shown in below Figure 1. The linear bearing is fixed on the bracket and the transducer base is fixed on the slotted link. The working principle is as follows: the motor drives the crank to rotate, so that the circular slider moves in the slotted link. The linear bearing restricts the movement of the slotted link to linear reciprocating motion, to realize the linear scanning of the transducer base. The mechanical scanning system has a scanning stroke range of 14 mm and a scanning speed of 168 mm/s. Therefore, the imaging frame rate of the system is 12 fps.

A 1000 P/R encoder (E6B2-CWZ6C, Omron) is used to record the rotation position of the motor in real time to obtain the precise positioning of the base, and the theoretical positioning accuracy of the system can reach 28 μm. The fabricated mechanical scanning system is shown in Figure 2. However, it should be noted that the mechanical scanning motion is nonlinear, which will affect the accuracy of high-frequency ultrasound imaging, so this paper adopts a motion compensation method to achieve corrected high-resolution imaging. See the Methods section for a detailed discussion.

#### 2.1.2. High-Frequency Ultrasound System

The proposed single-channel high-frequency ultrasonic imaging system is shown in Figure 3, mainly including the transmitting module, analog front end (AFE), acquisition module, and FPGA control module.

In the transmitting module, the trigger pulse generated by the inductor and MOSFET jointly generates a high-voltage pulse, and the high-voltage pulse excitation that can be used to drive the high-frequency transducer is formed through the matching circuit. The excitation waveform and waveform bandwidth are shown in Figure 4.

Since the attenuation of ultrasound in soft tissues increases significantly with the increase of ultrasonic frequency, the AFE of the system uses a multi-stage amplification gain block. The AFE contains a pre-amplifier, post-amplifier, band pass filter, and time gain amplifier (TGC). The pre-amplifier is placed near the transducer to reduce the signal attenuation caused by the long transmission wire. The pre-amplifier uses an HBT-based wideband low noise amplifier and an ultralow noise (0.9 nV/√Hz) and distortion voltage feedback op amp AD8099 cascades to improve the gain multiple and drive capability. By adjusting the gain factor and impedance matching of the AD8099, the amplitude range of the back-end input signal can be adjusted. The TGC is implemented using a 45 dB variable gain amplifier. TGC can control the amplification of time gain through the gain control signal at the DA output.

The filter adopts a band-pass filter to prevent the aliasing of high-frequency signals due to data acquisition. The filter has very good band-pass filtering performance, and the measured filter insertion loss value is shown in Figure 5. Its −3 dB passband signal is 28 MHz~84 MHz, which meets the system signal filtering requirements.

The acquisition module uses the ADC sampling chip for signal acquisition, and its acquisition frequency is 210 Msps@16-bit. The developed single-channel high-frequency ultrasound imaging system is shown in Figure 6. The system is controlled via an FPGA and uses the PCIE 3.0 × 4 interface for data transmission with the host computer.

#### 2.1.3. High-Frequency Transducer

A small high-frequency transducer was prototyped for the mechanical scanning ultrasound imaging system, just as shown in Figure 7. Considering the mounting fixation and imaging performance, the transducer size was designed to be 0.55 × 0.45 mm^2^ with a center frequency of 60 MHz. The transducer consists of a 30 μm PZT piezoelectric layer, a 0.6 mm silver epoxy backing layer, and a thin matching layer. The fabrication process is detailed in our previous studies [11,12]. Through a 46 AWG coaxial cable, the transducer was connected to the system.

The time domain pulse-echo response and normalized frequency spectrum of the fabricated high-frequency transducer were measured with a DPR500 pulser-receiver (JSR Ultrasonics, Pittsford, NY, USA). The received echo was reflected from an Ackley plastic plate placed 3 mm away from the transducer. Figure 8 displays the measured pulse-echo waveform (black line), and the frequency spectrum (blue line) was calculated by performing a Fourier transform on the echo data of the fabricated transducer. According to the measurement, the center frequency fc of the transducer is about 60.1 MHz, and the −6 dB bandwidth (BW) is as wide as 44.6%.

### 2.2. Method

#### 2.2.1. Mechanical Scanning Motion Compensation

Since the mechanical scanning of the system is not in uniform motion, the displacement distance at each moment is different when it rotates in one round. Take 1000 sampling points in one round according to the interval of the encoder, and their motion trajectory is shown in Figure 9 below. The trajectory as a whole presents a similar sinusoidal distribution, with the middle area being flatter and uniform, but running slower at both ends of the scan.

Firstly, as shown in Figure 10, the block diagram of the motion compensation, the mechanical structure was analyzed and calculated to obtain the motion characteristic curve of the transducer. Then, the position information of the transducer during the scanning and imaging process can be obtained, and next use the position information of the transducer to calculate the imaging results of different positions. Finally, the imaging results are processed by image processing, including mean filtering and median filtering, and other denoising processing.

#### 2.2.2. High-Speed Mechanical Scanning Imaging

The system drives the stepper motor to rotate at the rotation speed of 360 RPM, which can achieve an imaging speed of 12 fps. The rotating structure drives the encoder to rotate synchronously, generating a position-coded trigger signal. The encoder line speed directly determines the ultrasonic emission frequency of the imaging system, corresponding to the system’s lateral scanning density. Therefore, increasing the P/R number of rotary encoders is beneficial to improve the scanning accuracy of the system and reduce the errors caused by later position calibration. The system uses a 1000 P/R encoder, that is, there are 500 ultrasound transceivers in a single mechanical scanning process.

The proposed system works as shown in Figure 11. When the system is working, phase A of the encoder is used to synchronously trigger the pulse transmitter, acquisition module, and TGC to transmit high-frequency pulse, collect echo signals, and adjust the gain, respectively. The phase Z of the encoder is used as the starting signal of the acquisition module. When the phase Z signal is captured to start one frame signal acquisition, the second frame signal will start after capturing 500 times phase A signal, and then be continuously acquired. The received ultrasonic echo signal is staged in the memory of the acquisition card as RF data after preamplification, post-amplification, band pass filtering, and ADC data acquisition. The host computer reads the RF data through the PCIE interface, realizes the image data evenly spaced after interpolation through motion compensation, and completes the real-time imaging process through the CUDA acceleration algorithm.

## 3. Results

### 3.1. Wire Phantom Test

First, three 10 µm tungsten targets (Figure 12a) with a pitch of 0.5 mm were fabricated to assess the system resolution performance. The imaging results are shown in Figure 12b over a 30 dB dynamic range, indicating that the three wire targets are completely separated.

And as shown in Figure 13a, the RF data (black line) and their envelope signal (red line) were used to evaluate the resolution quantitative. Figure 13b,c show the resolution as demonstrated by evaluating the line spread function based on Rayleigh Criterion [13,14], where the axial resolution and lateral resolution can be calculated as 51 and 211 μm, respectively.

### 3.2. In Vitro Tissue Imaging of a Fish Eye

Lastly, in vivo imaging of a fish eye is conducted to verify the high imaging resolution, high SNR, and fast scanning of the developed system. All animal experimental procedures satisfy the laboratory animal protocol admitted by the animal experimentation ethics committee of Suzhou Institute of Biomedical Engineering and Technology. A volumetric ultrasound image with the field of view (i.e., 14 mm × 8 mm along the X and Y axes, respectively) is real-time displayed. As the ultrasound frequency increases, finer imaging target structures can be obtained. As shown in Figure 14, the cornea (C), iris (I), skin (S), and sclera and choroid (SC) structures are clearly visible. Specifically, the lamellar structure of the cornea can be clearly distinguished. During the imaging process, motion compensation processing can obtain the accurate position of the transducer; after calibration, high position accuracy can significantly improve the resolution and SNR of the image through aperture synthesis method, so that the virtual source synthetic aperture technology can be used to effectively suppress noise interference [15], thereby improving the imaging quality. Compared to the image before motion compensation, the compensated image reconstructs the curvilinear features of the fisheyel; at the same time, the graphic distortion of the ciliary body part is effectively resolved.

## 4. Discussion

A high-frequency ultrasound imaging system based on mechanical scanning was developed. The design and imaging method of the system was described in detail. The system was used to image the tungsten wire targets and the fisheye structure. During the experiment, the fisheye was fixed on a stage in the water, and the mechanical scanning probe was placed 2 mm above the fisheye for reciprocating scanning. The experimental results show that the proposed method can achieve high-frequency ultrasonic scanning imaging without increasing the complexity and cost of the system, and the imaging resolution and quality can reach the desired level. This system offers a compact and affordable solution for high-frequency ultrasound imaging, such as ophthalmic ultrasound, skin ultrasound, ultrasound-guided catheterization, and so on, because of its low complexity, low cost, and high resolution compared to array-based US imaging. In addition, just as the clinical ultrasound system can be equipped with multiple probes to meet the detection of different tissues, in fact, this proposed ultrasound imaging system could also be changed by the mechanical scanning transducer, such as using 20 MHz or 30 MHz transducers, to take into account the imaging resolution and imaging depth needs of different tissues and applications. The mechanical structure and motor could be improved to increase the imaging frame rate. Additionally, the probe will be fully encapsulated for handheld and convenient use. Furthermore, our proposed system can also be applied to high-resolution ultrasound imaging of small animals, such as mice, rabbits, etc. Therefore, we believe that the system has a very broad clinical application and application prospects.

## Figures and Tables

**Figure 1 biosensors-13-00032-f001:**
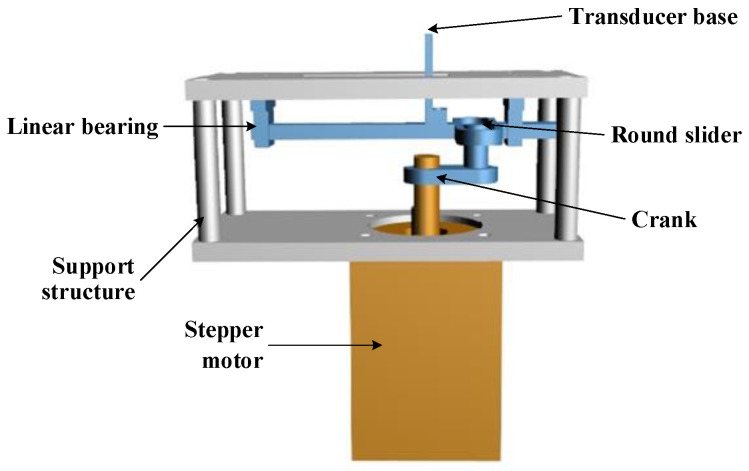
Design drawing of the mechanical scanning system.

**Figure 2 biosensors-13-00032-f002:**
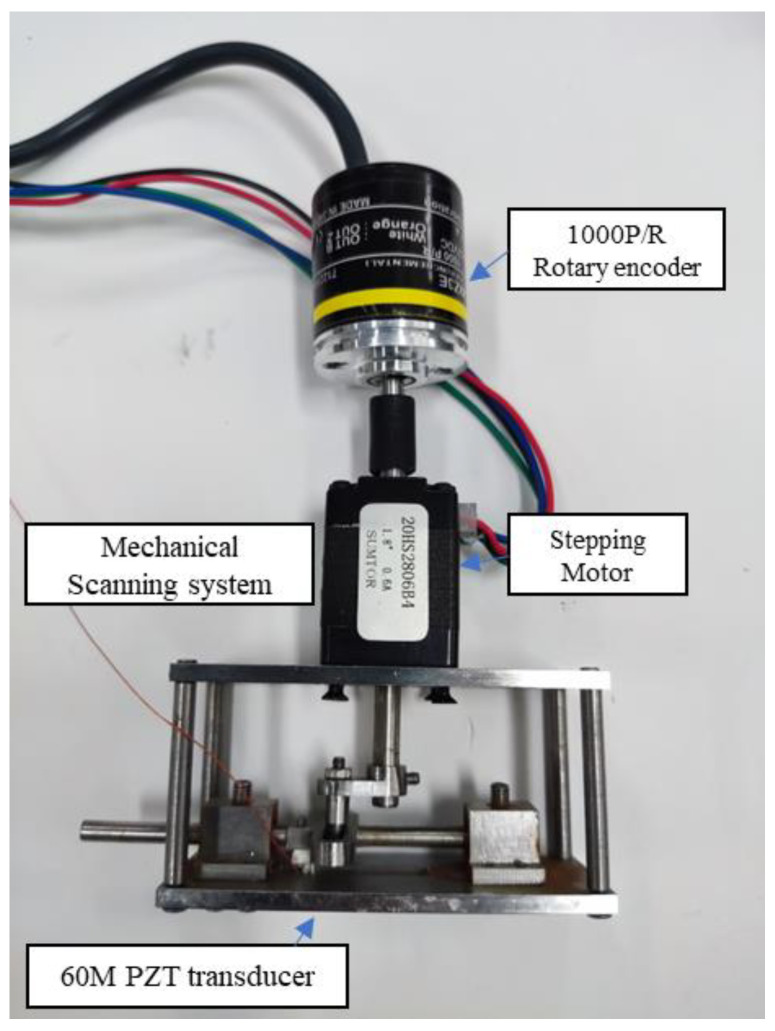
The fabricated mechanical scanning system.

**Figure 3 biosensors-13-00032-f003:**
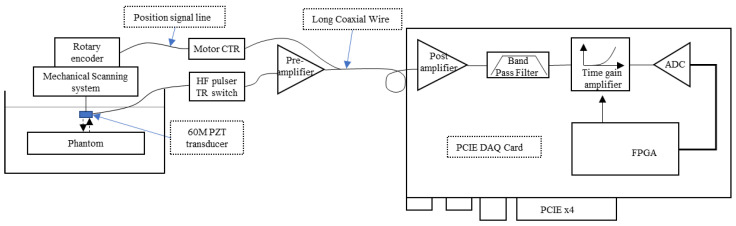
Schematic diagram of high frequency ultrasound imaging system.

**Figure 4 biosensors-13-00032-f004:**
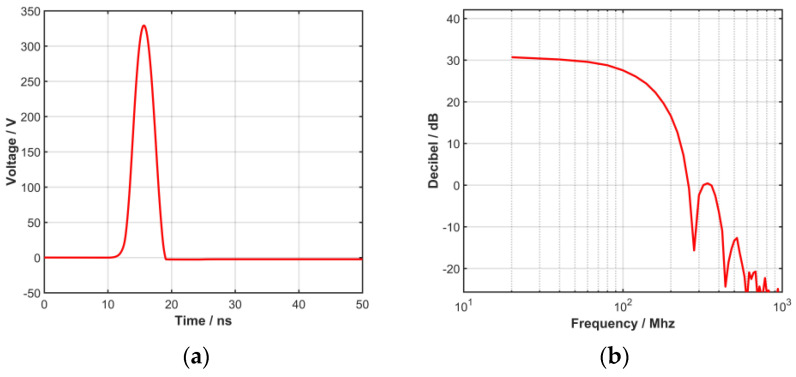
(**a**) The generated high voltage pulse and (**b**) bandwidth.

**Figure 5 biosensors-13-00032-f005:**
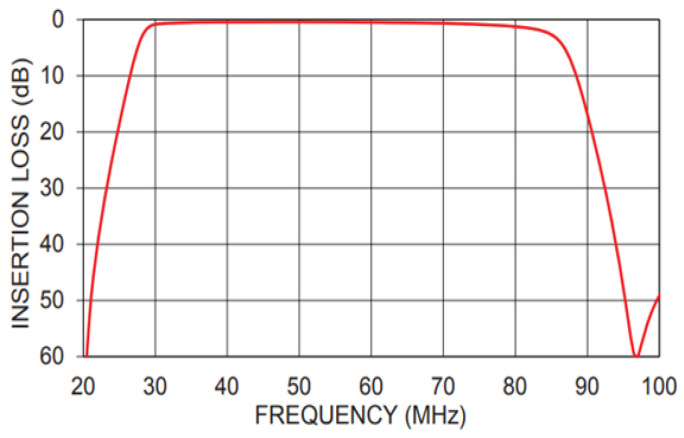
The measured filter insertion loss value curve.

**Figure 6 biosensors-13-00032-f006:**
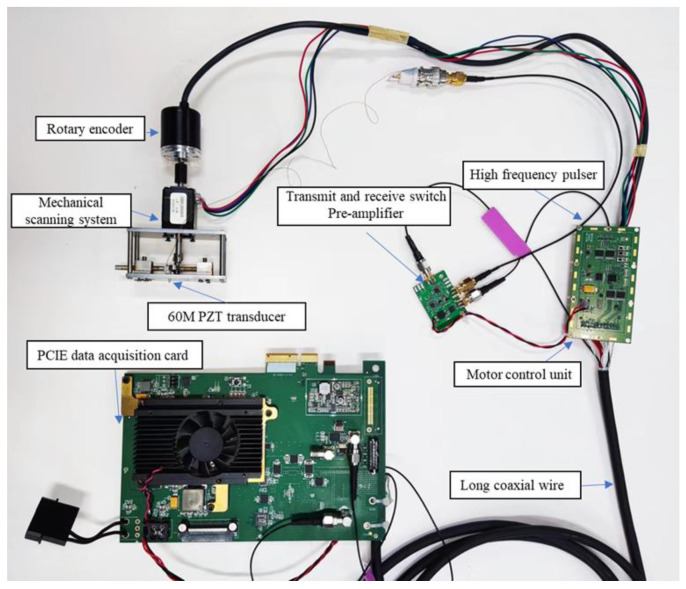
The developed single-channel high-frequency ultrasound imaging system.

**Figure 7 biosensors-13-00032-f007:**
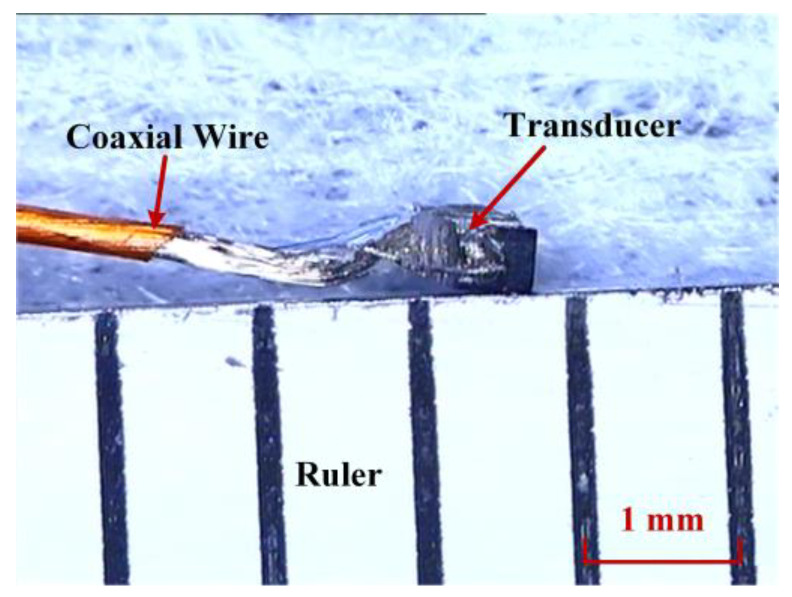
The fabricated small high frequency ultrasonic transducer.

**Figure 8 biosensors-13-00032-f008:**
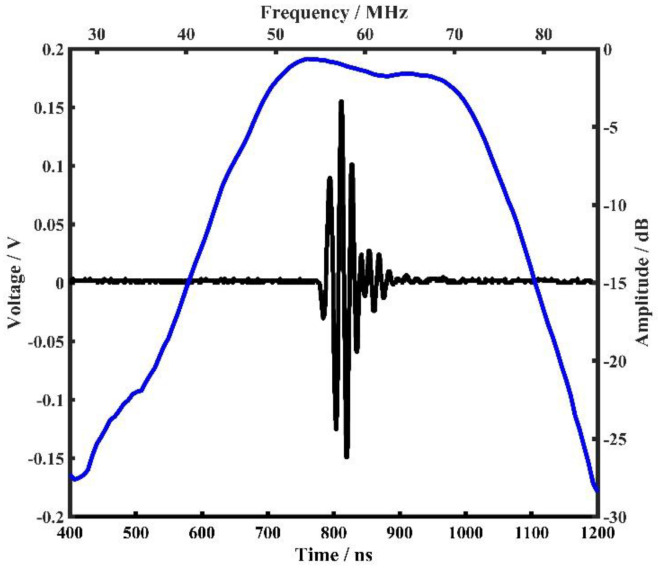
The measured pulse-echo response (black line) and calculated frequency spectrum (blue line) of the fabricated high-frequency transducer.

**Figure 9 biosensors-13-00032-f009:**
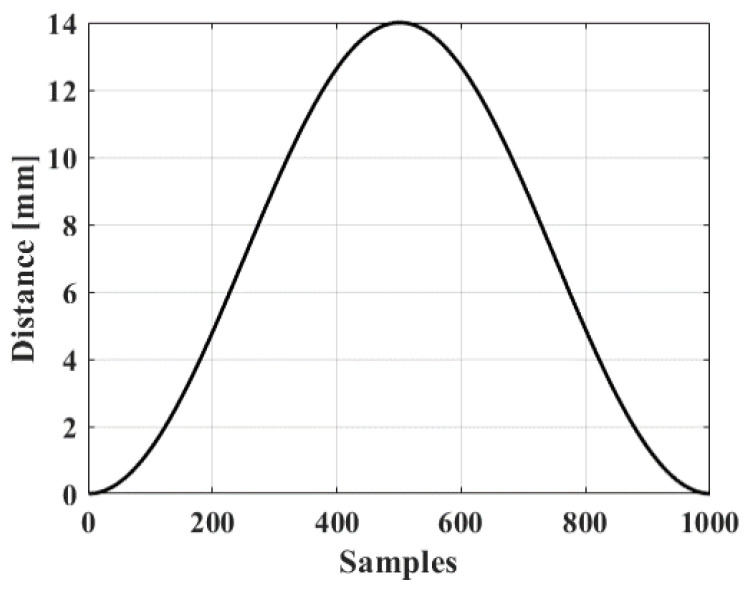
Mechanical scanning motion curve simulation diagram in one round with 1000 samples.

**Figure 10 biosensors-13-00032-f010:**
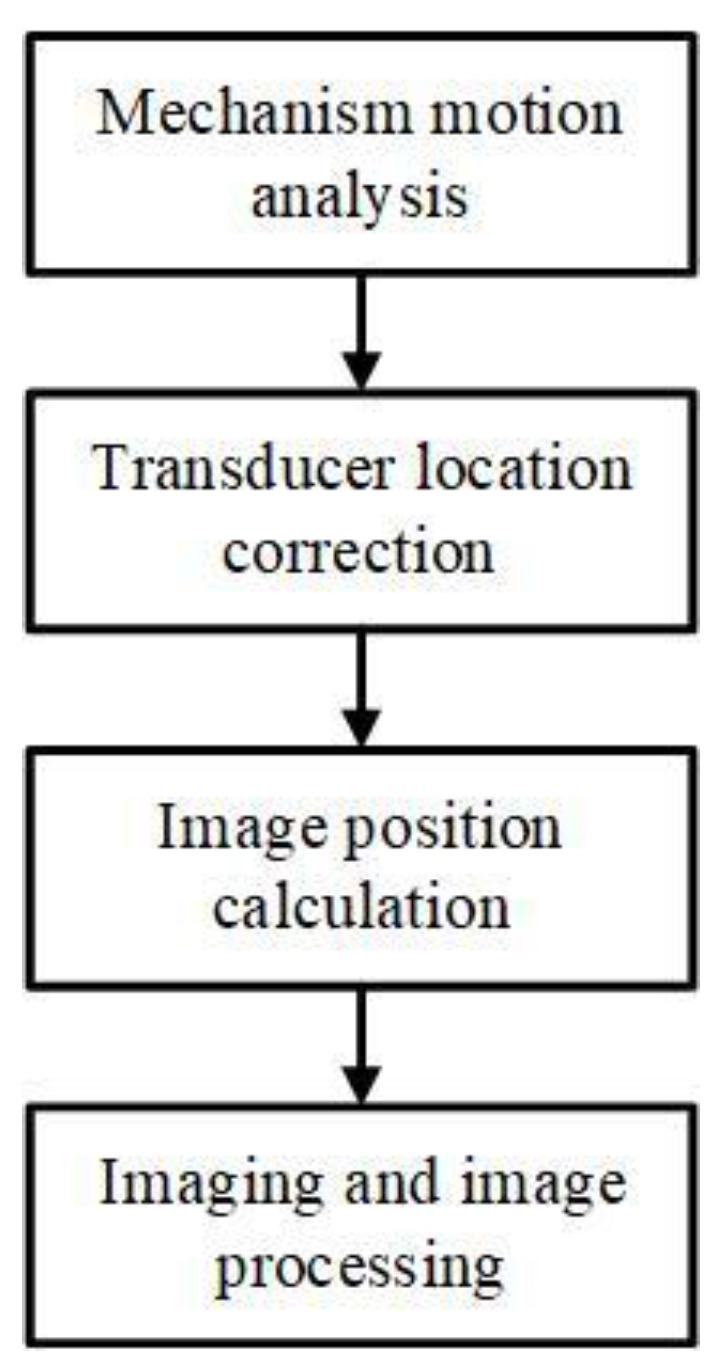
The block diagram of the motion compensation.

**Figure 11 biosensors-13-00032-f011:**
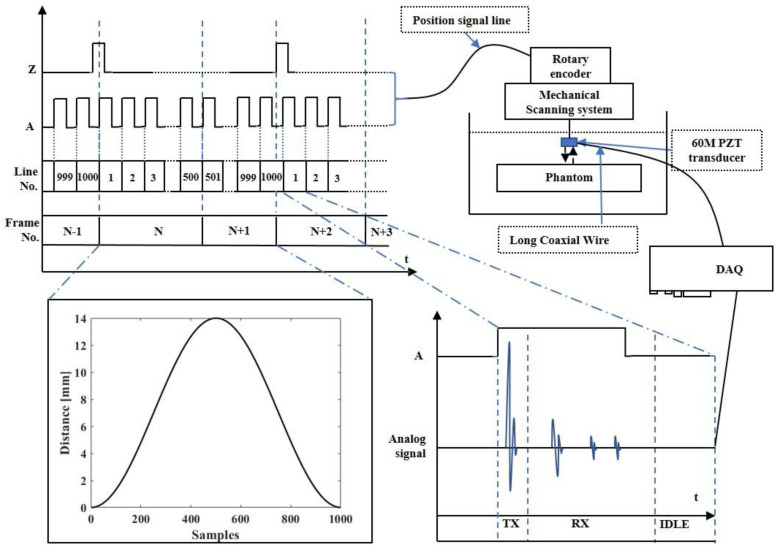
Workflow diagram of the proposed system.

**Figure 12 biosensors-13-00032-f012:**
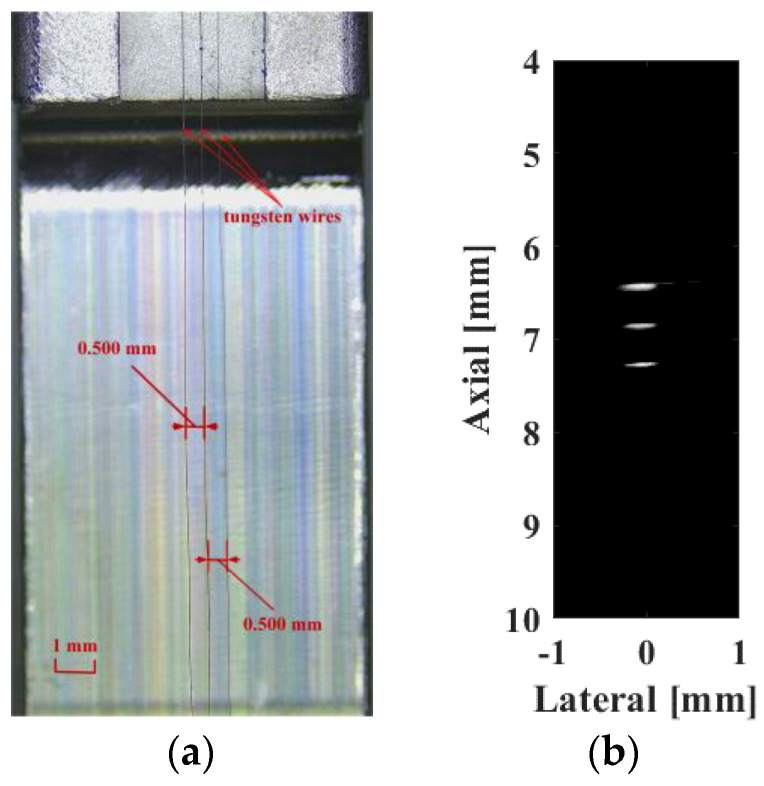
(**a**) The tungsten wire phantom and (**b**) system-captured wire phantom image.

**Figure 13 biosensors-13-00032-f013:**
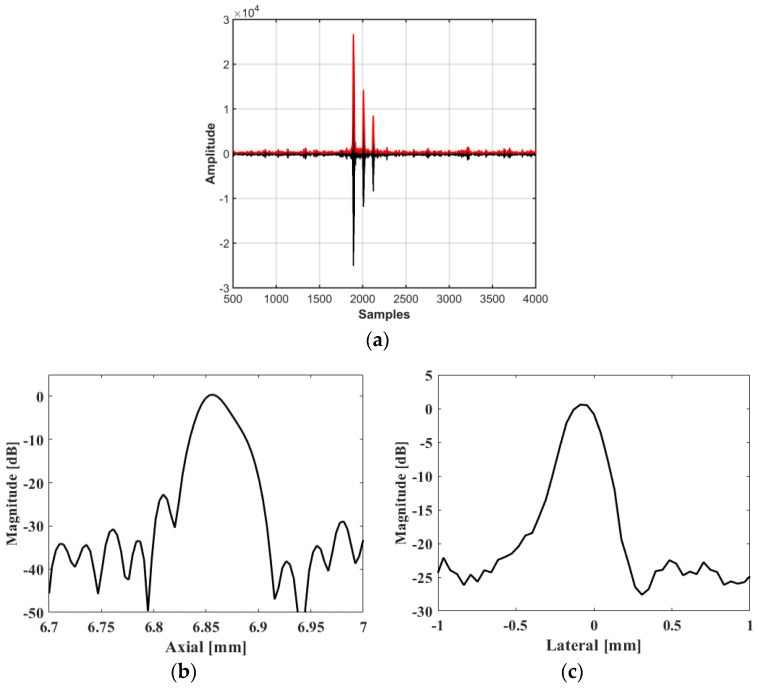
(**a**) The captured raw data and their envelope. (**b**) Axial beam profiles and (**c**) lateral beam profiles.

**Figure 14 biosensors-13-00032-f014:**
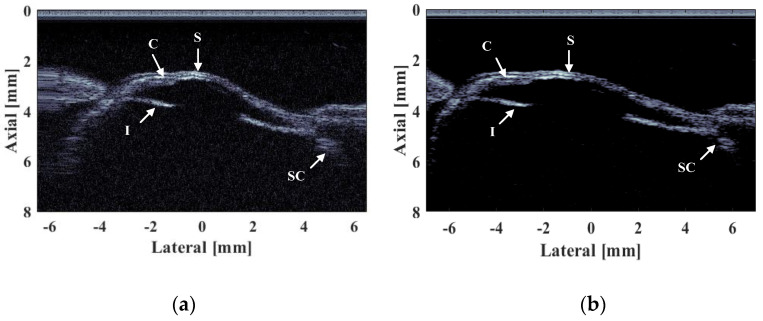
In vivo ultrasound imaging of a fisheye: (**a**) Raw image, (**b**) motion compensated image. Dynamic range: 45 dB.

## Data Availability

The data supporting this study’s findings are available in the Database Science Data Bank (DOI:10.6084/m9.figshare.13279820.V1).

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
