# Peer review of "A High-Frequency Mechanical Scanning Ultrasound Imaging System"

_biosensors, 2022, doi:10.3390/bios13010032_

Round 1

Reviewer 1 Report

In this manuscript a high frequency ultrasound imaging system based on mechanical scanning was proposed. This system is explained in detail. It seems to be a very effective system in clinical use. It is appropriate to publish the article in the journal after some revisions.

1. What are the differences of the USG system mentioned in the article with the existing systems? What is its advantage over others?

2. Especially for which tissues is this recommended system more effective? Especially clinical use and differences with other existing systems should be discussed.

Reviewer 2 Report

In this manuscript, the author applied reciprocating feed mechanism for ultrasound imaging scanning. A n imaging range of 14 mm and a frame rate of 12 fps were achieved. In vivo experiment was performed to demonstrate the performance. Overall, this manuscript is well written. I will recommend accepting this paper after addressing the comments below.

1.      A mechanical scanning motion compensation have been performed to correct the uneven scanning. What is the reliability of this method? Does the compensation need redo after a certain time? What is positioning accuracy after correction?

2.      For current system, 1D scanning have been achieved, what is challenging for 2D scanning?

3.      Figure 12(a) is not clear, the labels are not recognized.

4.      Please provide full name of all the abbreviations shown in Figure 6 in the caption.

6.      Please add a scale bar in Figure 7.

7.      With motion compensation, beside the improved resolution, it seems that the image SNR was improved as well. In Figure 14(a), background looks much nosier than Figure 14(b).

Please provide all the abbreviation in the captions.

8.      A brief discussion on the limitation of current design and next step will be helpful.

Round 2

Reviewer 1 Report

The article with its current form can be published in the journal. It was quite interesting study. I think that this manuscript will make important contributions to the literature.

Best regards.